# Association between Polymorphism rs1799732 of *DRD2* Dopamine Receptor Gene and Personality Traits among Cannabis Dependency

**DOI:** 10.3390/ijerph191710915

**Published:** 2022-09-01

**Authors:** Jolanta Chmielowiec, Krzysztof Chmielowiec, Jolanta Masiak, Małgorzata Śmiarowska, Aleksandra Strońska-Pluta, Violetta Dziedziejko, Anna Grzywacz

**Affiliations:** 1Department of Hygiene and Epidemiology, Collegium Medicum, University of Zielona Gora, 28 Zyty St., 65-046 Zielona Gora, Poland; 2Second Department of Psychiatry and Psychiatric Rehabilitation, Medical University of Lublin, 1 Głuska St., 20-059 Lublin, Poland; 3Department of Pharmacokinetics and Therapeutic Drug Monitoring, Pomeranian Medical University, 70-111 Szczecin, Poland; 4Independent Laboratory of Health Promotion, Pomeranian Medical University in Szczecin, 11 Chlapowskiego St., 70-204 Szczecin, Poland; 5Department of Biochemistry and Medical Chemistry, Pomeranian Medical University, 70-111 Szczecin, Poland

**Keywords:** addiction, *DRD2* gene

## Abstract

Compared to other addictive substances, patients with cannabis addiction are significantly outnumbered by those who report dependence on other, more addictive substances. Unfortunately, most cannabis addiction goes untreated, and among those who choose treatment, the requirements are much higher for adolescents and young adults. The aim of the study: To examine the relationship of cannabinoid dependency in the genetic context—the association between the rs1799732 polymorphism of the *DRD2* gene and psychological traits and anxiety. Methods: The study group consisted of 515 male volunteers. Of these, 214 patients were diagnosed with cannabis addiction and 301 were non-addicted. Patients were diagnosed with NEO Five-Factor Personality Inventory (NEO-FFI), and State–Trait Anxiety Inventory (STAI) questionnaires. The interactions between personality traits and polymorphisms in the *DRD2* rs1799732 gene were investigated in a group of cannabis-addicted patients and non-addicted controls using the real-time PCR method. Results: Compared to the control group, the case group obtained significantly higher scores on the STAI State, STAI Trait, Neuroticism and Openness scales, as well as lower scores on the Extraversion, Agreeableness, and Conscientiousness scales. There was no statistically significant difference between addicts and the control group in the frequency of genotypes, but there was a statistically significant difference between addicts and the control group in the frequency of the *DRD2* allele rs179973. The multivariate ANOVA analysis showed a statistically significant influence of the *DRD2* rs1799732 genotype on the NEO-FFI agreeableness scale and a statistically significant effect of addiction to cannabinoids or its absence on the NEO-FFI agreeableness scale score. Conclusions: Studying homogeneous subgroups—as in our study—seems reasonable, particularly when combined with genetic determinants and psychological traits. In multigenic and multifactorial entities, such a strategy has a future.

## 1. Introduction

The most commonly used illegal drug in Western societies is marijuana (Cannabis sativa) [1,2,3,4]. According to the United Nations, in 2018, as many as 192 million people, or 3.9% of the world’s adult population, used cannabis in the preceding year [5]. Cannabis use disorder (CUD) is the inability to stop using cannabis despite the physical and psychological damage. According to data from 2016, as many as 22.1 million people met the CUD diagnostic criteria, and two-thirds were men [6,7,8,9]. CUD is one of the most common SUDs.

Despite the widespread use of cannabis, only the group of adolescents (25.4%) and young adults (19.0%) who use cannabis have seen a development of abuse or even addiction [10]. As many as 10% of cannabis users become daily users [11]. Day-to-day use is the best predictor of CUD, and as many as one-third develop an addiction. Although, compared to other drugs cannabis addiction occurs after prolonged use; people at a young age are more likely to become dependent on cannabis [12].

Of the neurotransmitter systems linked with addiction, dopamine has received the most attention, given its strong role in reward, motivation, and goal-directed behavior. Similarly to other substances of abuse, acute THC increases dopamine release in the striatum of healthy subjects [13]. Following chronic use, there is a reduction in stimulated dopamine levels in CUD [14], as in other SUDs (e.g., psychostimulants, nicotine, alcohol, and opioids). Early age of onset or longer duration of cannabis use correlates with reduced stimulated striatal dopamine release (evoked by psychostimulant administration) [15]. The lower striatal dopamine release apparent in heavy cannabis users relates to inattention and more significant negative symptoms [16], and it inversely correlates with negative emotionality and addiction severity [17]. The reduced release also corresponds with decreased dopamine synthesis in cannabis-dependent individuals [18], an effect associated with greater apathy [19,20].

Compared to other addictive substances, patients with cannabis addiction are significantly outnumbered by those who report dependence on other, more addictive substances [21]. Unfortunately, most cannabis addiction goes untreated, and among those who choose treatment, the requirements are much higher for adolescents and young adults [11]. Annual remission is around 17% among those not seeking treatment [22]. Early onset and intensity of use, early life trauma, mental health problems, and genetic susceptibility play an essential role in the development and severity of CUD [23,24,25].

In our study, it is for this reason that we decided to investigate the problem, but with a special focus on genetic determinants and personality traits. The problem is still huge, and despite numerous studies, we do not yet know the specific biological and psychological determinants (particularly when analyzed simultaneously). The dopamine receptor gene *DRD2* was selected as the genetic factor. The dopamine D2 receptor gene is located on chromosome 11q23 and covers an area of 65.56 kb. Eight coding regions (exons) within the gene are transcribed into 2713 kb mRNA. As a result of translation, the protein D2 receptor is 443 amino acids in size. As a result of the exclusion of exon six at the stage of transcription, a receptor variant is created that is 29 amino acids shorter [26]. The research results indicate at least a partial role of polymorphisms in the regions not coding the *DRD2* gene in shaping the risk of developing drug addiction throughout life [27]. 

One DRD2 gene polymorphism that has been investigated extensively is a cytosine (C) insertion/deletion (Ins/Del) at nucleotide position −141C (−141C Ins/Del, rs1799732) in the promoter region, which may regulate DRD2 transcription by modulating the binding of transcription factors [28]. Arinami et al. [28] show that variation in the genomic sequence of the promoter region of the D2 receptor gene (DRD2) could affect the expression or regulation of the gene. The DRD2 5′-promoter fragments drove the transcription of heterologous luciferase constructs in the Y79 cell line expressing DRD2 as well as in DRD2 non-expressing 293 cells. The fragment that contained the −141C Del allele showed a decrease in promoter strength as compared with the fragment that contained the −141CIns allele in Y-79 and 293 cells. The position of the polymorphism is part of a putative binding site for the transcription factor Sp-1, 5′-CCAGGCCGGGGATCGCC.

In an in vivo study, Jönsson et al. [29] found a significant association between the presence of a putative functional DRD2 promoter allele (−141C Del) and high striatal DA receptor density in healthy subjects. These studies suggested that the rs1799732 polymorphism may be involved in the regulation of DRD2 expression (mRNA level and/or protein level).

Previous publications have described the relationship of the rs1799732 polymorphism of DRD2 with nicotine [30], alcohol addiction [31,32,33,34], and opioid dependence [35]. Therefore, in our research, we decided to examine the relationship of the rs1799732 polymorphism of DRD2 with cannabis dependency and with personality traits.

Personality traits were an additional factor that was analyzed in our study. Personality affects many aspects, such as behavior, lifestyle, and maintenance of normal functions throughout life. The Big Five-factor model is most often used in personality research [36,37,38]. It consists of five traits: Openness, Conscientiousness, Extraversion, Agreeableness, and Neuroticism. Human differences are determined by these characteristics and related to behavior, emotions, motivation, and cognition [39]. Higher levels of anxiety are associated with susceptibility to substance addiction. Several studies describe the correlation of addiction with drug traits as measured by the STAI questionnaire [40]. The NEO Personality Inventory (NEO-FFI) is most often used to analyze personality traits [38]. The State and Trait Anxiety Inventory (STAI) is another tool used in addiction research. It allows to measure both the state, the reaction to a given situation, and the trait, the general tendency to react in a certain way in a state of anxiety [41]. As described above, both genetics and personality are factors that influence addiction. Therefore, our current study aimed to evaluate the effects of both of these components on cannabis use and the influence of the interaction of certain genetic variants with personality traits and anxiety. The analysis was carried out by comparing the polymorphism in the *DRD2* gene and personality traits measured by the Big Five Questionnaire (NEO FFI) and anxiety measured by the Anxiety Trait Inventory (STAI) in two study groups—cannabis-dependent and non-addicted.

## 2. Materials and Methods

### 2.1. Materials

The study group consisted of 515 male volunteers. Of these, 214 patients were diagnosed with cannabis addiction (mean age = 27.46, SD = 6.12) and 301 were non-addicted (mean age = 22.14, SD = 4.57). The Bioethics Committee previously approved the study of the Pomeranian Medical University in Szczecin (KB-0012/106/16). All participants gave their written consent to participate in the study, and the studies were conducted in the Independent Health Promotion Laboratory. Addicts were recruited after at least three months of abstinence in addiction treatment centers. Patients diagnosed with polysubstance use disorder with a depressive episode and the control group were examined by a psychiatrist using the Mini International Neuropsychiatric Interview (MINI), NEO Five-Factor Personality Inventory (NEO-FFI), and State–Trait Anxiety Inventory (STAI) questionnaires.

The interactions between personality traits and polymorphisms in the *DRD2* rs1799732 gene were investigated in a group of cannabis-addicted patients and non-addicted controls.

### 2.2. Measures 

MINI-International Neuropsychiatric Interview is a structured diagnostic interview designed to evaluate the diagnoses of psychiatric patients according to the DSM-IV and ICD-10 criteria.

STAI measures anxiety as a trait of anxiety (trait A), which can be described as a persistent predisposition to having worries, stress, and discomfort, and anxiety states (A-states) such as anxiety, fear, and momentary stimulation of the autonomic nervous system in response to specific situations.

The Personality Inventory (NEO-FFI Five-Factor Inventory, NEO-FFI) contains six components for each of the five traits—neuroticism (anxiety, hostility, depression, self-awareness, impulsivity, susceptibility to stress), extroversion (warmth, sociability, assertiveness, activity, emotion seeking, positive emotions), openness to experience (fantasy, aesthetics, feelings, actions, ideas, values), agreeableness (trust, straightforwardness, altruism, compliance, modesty, tenderness), and conscientiousness (competence, order, duty, striving for achievements, self-discipline, consideration) [38]. 

The results of both inventories, i.e., NEO-FFI and STAI, were reported as the sten scores. The conversion of the raw score to the sten scale was carried out following the Polish standards for adults, where it was assumed that 1–2 corresponded to very low results; 3–4 was responsible for low results, 5–6 was responsible for average results; 7–8 was responsible for high results; and 9–10 sten was responsible for very high results.

### 2.3. Genotyping

The genomic DNA was isolated from venous blood by using standard procedures. Genotyping was conducted with the real-time PCR method. Details have been described previously [41]. The fluorescence signal was plotted as a function of temperature to provide melting curves for each sample. The *DRD2* gene peaks are read (rs1799732), 56.64 °C for the C allele and 62.85 °C for the (-) allele. 

### 2.4. Statistical Analysis

A concordance between the genotype frequency distribution and Hardy–Weinberg equilibrium (HWE) was tested using the HWE software (https://wpcalc.com/en/equilibrium-hardy-weinberg/ (20 March 2021). The relations between *DRD2* gene (rs1799732) variants, Cannabis Dependency and control subjects, and the NEO Five-Factor Inventory were analyzed using a multivariate analysis of factor effects ANOVA [NEO-FFI/scale STAI × genetic feature × control and Cannabis Dependency × (genetic feature × control and Cannabis Dependency)]. The condition of homogeneity of variance was fulfilled (Levene test *p* > 0.05). The analyzed variables were not distributed normally. The NEO Five-Factor Inventory (Neuroticism, Extraversion, Openness, Agreeability, and Conscientiousness) was applied and compared using the U Mann–Whitney test. The *DRD2* gene (rs1799732) genotype frequencies between healthy control subjects and Cannabis Dependency were tested using the chi-square test. All computations were performed using STATISTICA 13 (Tibco Software Inc., Palo Alto, CA, USA) for Windows (Microsoft Corporation, Redmond, WA, USA).

## 3. Results

These frequency distributions were in accordance with the HWE both in the Cannabis Dependency and control subjects (Table 1).

No statistically significant differences were found in the frequency of *DRD2* rs1799732 genotypes in the tested cannabis addicts compared to the control group (del/del 0.03 vs. del/del 0.01, ins/ins 0.74 vs. ins/ins 0.80, ins/del 0.23 vs. ins/del 0.19, χ^2^ = 3,97, *p* = 0.138). However, statistically significant differences in the frequency of *DRD2* rs1799732 gene alleles were found between cannabinol-dependent subjects and the control group (del 0.15 vs. del 0.11, ins 0.85 vs. ins 0.89, χ^2^ = 3.87, *p* = 0.049) (Table 2).

The means and standard deviations for all the NEO-FFI results and the STAI scale state and trait scale variant interactions for the Cannabis Dependency and control subjects are presented in Table 3.

The test subjects addicted to cannabis compared to the control group obtained higher scores in the assessment of anxiety (STAI) as a state (5.82 vs. 4.69; Z = 5.418; *p* ≤ 0.000) and trait (7.10 vs. 5.16; Z = 8.619; *p* ≤ 0.000), NEO-FFI neuroticism scale (6.70 vs. 4.68; Z = 9.472; *p* ≤ 0.000), and NEO-FFI openness scale (5.04 vs. 4.53; Z = 2.835; *p* = 0.004). On the other hand, lower results were found for the NEO-FFI extraversion scale (5.66 vs. 6.37; Z = −3.667; *p* ≤ 0.000), the NEO-FFI agreeableness scale (4.29 vs. 5.60; Z = −6.825; *p* ≤ 0.000), and the NEO-FFI conscientiousness scale (5.49 vs. 6.08; Z = −2.845; *p* = 0.004). 

The results of the 2 × 3 factorial ANOVA of the NEO Five-Factor Personality Inventory (NEO–FFI) and the State–Trait Anxiety Inventory (STAI) sten scales are summarized in Table 4.

A significant statistical impact of cannabis dependence or absence and *DRD2* genotype rs1799732 was demonstrated for flux as a feature and score of the NEO-FFI agreeableness scale.

There was a statistically significant effect of *DRD2* genotype interaction rs1799732 and addiction to cannabis or its absence (control group) on the trait anxiety scale score (F_2.507_ = 4.39; *p* = 0.013; η^2^ = 0.017; Figure 1). The potency observed for this factor was 76%, and the polymorphism of the *DRD2* gene explained approximately 2% of rs1799732 and cannabis dependence or lack thereof on trait anxiety score variance. There was also a statistically significant effect of addiction to cannabis or its absence on the trait anxiety scale score (F_1.507_ = 27.02; *p* < 0.0001; η^2^ = 0.051). The potency observed for this factor was over 99% and approximately 5% was explained by cannabis dependence or lack thereof on the variance in the trait anxiety score. Table 5 shows the results of the post-hoc test.

A statistically significant influence of the *DRD2* genotype was demonstrated: rs1799732 on the NEO-FFI agreeableness scale score (F_2.507_ = 4.33; *p* = 0.013; η^2^ = 0.017). The potency observed for this factor was set at 75%, and approximately 2% was explained by the polymorphism of the *DRD2* gene rs1799732 variances of the NEO-FFI Agreeableness Scale Score. There was also a statistically significant effect of addiction to cannabis or its absence on the NEO-FFI agreeableness scale score (F_1.507_ = 17.50; *p* < 0.0001; η^2^ = 0.033). The potency observed for this factor was estimated to be approximately 99%, and approximately 3% was explained by cannabis dependence or lack thereof by variations in the NEO-FFI agreeableness score (Table 4).

## 4. Discussion

Addiction and abuse of illegal psychoactive substances are major problems in public health [42]. To develop a more effective approach to prevention and treatment, it is necessary to better understand the source of individual differences in risk. Extensive research suggests that genetic factors play an essential role in developing substance use disorder [42,43].

We aim to examine the combination of personality traits measured by two inventories—NEO-FFI and STAI—and genetic factors in the contextual occurrence of addiction.

We found many important correlations regarding the factors mentioned above. In addicts, the results of neuroticism and openness were higher, and the results of extraversion, agreeableness, and conscientiousness were lower than in the control group. Compared to the control group, the subjects addicted to cannabis had significantly higher scores on the assessment of anxiety (STAI) as a state and trait scale, neuroticism, openness, and lower scores on the extraversion, agreeableness, and conscientiousness scales. Additionally, the analysis of the results of the NEO-FFI inventory shows a significant effect of addiction to cannabis or its absence on the NEO-FFI agreeableness scale score. Our observations are congruent with the latest data that neuroticism, agreeableness, and conscientiousness are associated with drug use. It was stated that high neuroticism, high openness to experience, and low agreeableness may also be somewhat due to typical familiar effects not only as a result of the individual personality traits [44]. By these authors, it was found that high openness to experience was connected with cannabis use, high neuroticism was related to wrong drug prescription, while both high extraversion and low agreeableness concerned cocaine/crack and stimulant use [44].

There was no statistically significant difference in the frequency of *DRD2* rs1799732 genotypes in the tested cannabis addicts compared to the control group. However, we found statistically significant differences in the frequency of *DRD2* rs1799732 gene alleles between cannabis-dependent subjects and the control group. 

Multi-factor ANOVA of addicted subjects and control subjects and the *DRD2* rs1799732 variant interaction approximated the statistical significance for the STAI trait and agreeableness scale. 

As it has been shown in other data, neuroticism is thought to modulate the genetic risk to cannabis dependence and this range is estimated to be about 15,5–19,5% (9-fold increase). The SNPs (single nucleotide polymorphisms) of the *DRD2* and *PENK* (proenkephalin) genes seem to be candidates playing a significant role in this process [45]. These gathered data let us assume that low agreeableness together with high neuroticism might be the significant traits for cannabis addiction.

In patients with addiction to psychoactive substances, anxiety disorders often coexist and are more common in families with the problem of using psychoactive substances [46]. The anxiety–impulsive personality traits in people affected by substance use disorders and their families reach higher values than in the control group. Anxiety–impulsive personality traits are a potential endophenotype risk of developing cocaine or amphetamine addiction [47]. People with higher levels of anxiety are more prone to substance addiction. Research confirms the relationship between the anxiety traits measured by STAI and dependence [48]. Addicted patients obtained a higher score on the STAI inventory and the depression scale and lower on the tolerance scale (stress tolerance). Dealing with stress and negative mood states is a common theme of substance use among heavy addicts [49].

Research shows that personality traits play a leading role in problematic substance use. Substance abusers and non-drug abusers have higher measures of stress sensitivity than controls, suggesting that neuroticism may be an endophenotype in substance use disorders. A study by Terracciano et al. [50] shows that low conscientiousness scores and high neuroticism scores indicate an association with many psychoactive substances such as tobacco, heroin, and cocaine. Cannabis users score low on the conscientiousness scale but average on the neuroticism scale and high on the openness scale, which is the hallmark of cannabis users. 

This observation shows that there are two extremes of the personality characteristics which may be involved in CUD—on one side is the high neurotic predisposition and the other is schizothymic construct. Soler et al. [51] proved that the cannabis-addicted patients with the genetic variant of *ZNF804* (rs1344706) were characterized by their significant relationship with schizotypal personality traits and psychosis proneness. Furthermore, the schizotypy scores were positively correlated with the cannabis use pattern (dose and frequency) in these subjects. Although, a large meta- analysis performed in 24 studies with 6075 cases and 6643 controls involved with the rs 1799732 *DRD2* variant indicated no association of this locus with schizophrenia [52]. The data collected by means of functional magnetic resonance gave evidence for the role of the *DRD2* rs 1799732 polymorphism in response inhibition and the self-monitoring process in impulsive behavior as is seen alcohol abuse disease [53].

According to literature reports, personality traits may become a predisposing factor to addiction. The most widely described trait is impulsivity, which Barrat defines as “acting under the pressure of the moment”. Experimenting with different substances and behaviors that can lead to addiction is the decision for addicted patients. There is a clear connection between this trait with various addiction profiles. Compared to alcohol addicts, drug addicts are more impulsive [54].

Dopaminergic transmission is related to novelty seeking, a feature related to addiction and relapse [55,56,57,58,59]. Additionally, extraversion is a personality trait associated with the dopaminergic system. Studies of twins reveal that genetics have a different effect on each trait—from 25% to 61% [60].

Dopaminergic conductivity plays a key role in shaping the reward phenomenon in response to psychoactive substances. Despite the use of various methods and different groups of patients, the research results indicate a certain role of *DRD2* gene polymorphisms in addiction [61].

## 5. Conclusions

Compared to the control group, the case group obtained significantly higher scores on the STAI State, STAI Trait, Neuroticism, and Openness scales and lower scores on the Extraversion, Agreeableness, and Conscientiousness scales. There was no statistically significant difference between addicts and the control group in the frequency of genotypes, but there was a statistically significant difference between addicts and the control group in the frequency of the *DRD2* allele rs1799732.

The multivariate ANOVA analysis showed a statistically significant influence of the *DRD2* rs1799732 genotype on the NEO-FFI agreeableness scale and a statistically significant effect of addiction to cannabis or its absence on the NEO-FFI agreeableness scale score.

The multi-factor ANOVA of addicted subjects and control subjects and the *DRD2* variant interaction approximated the statistical significance for the STAI trait.

We are careful in our conclusions because we still know too little about the biological determinants of addiction. Studying homogeneous subgroups—as in our study—seems reasonable, particularly when combined with genetic determinants and psychological traits. In multigenic and multifactorial entities, such a strategy has a future.

## Figures and Tables

**Figure 1 ijerph-19-10915-f001:**
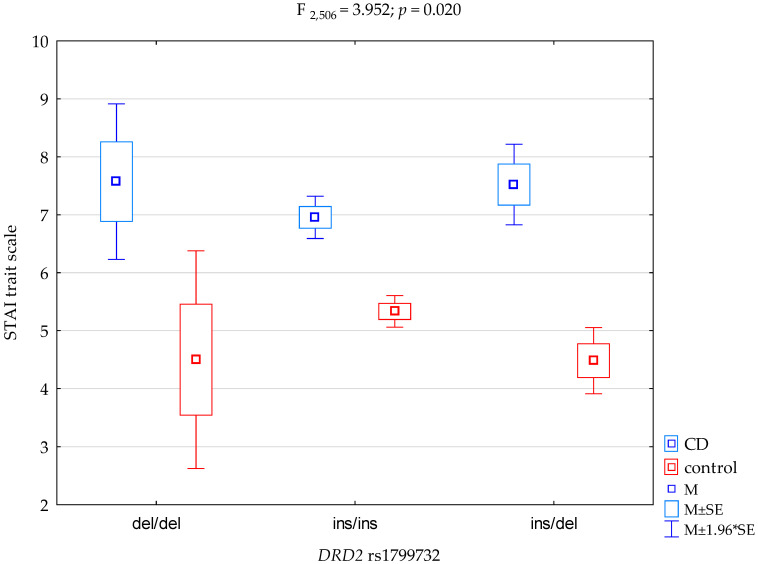
Interaction between the patients diagnosed with polysubstance use disorder comorbid with cannabinoid dependence (CD)/control and the *DRD2* gene rs1799732 and STAI trait scale.

**Table 1 ijerph-19-10915-t001:** Hardy–Weinberg’s law for patients diagnosed with polysubstance use disorder comorbid with a depressive episode Cannabis Dependency and control subjects.

Hardy–Weinberg Equilibrium Calculator, Including Analysis for Ascertainment Bias		Observed (Expected)	Allele Freq	χ^2^ (*p* Value)
*DRD2* rs1799732 Cannabis Dependency n = 214	ins/ins	158 (155.64)	p (ins) = 0.853 q (del) = 0.147	1.656 (>0.05)
	ins/del	49 (53.73)		
	del/del	7 (4.64)		
*DRD2* rs1799732 control n = 301	ins/ins	241 (240.40)	p (ins) = 0.894 q (del) = 0.106	0.132 (>0.05)
	ins/del	56 (57.20)		
	del/del	4 (3.40)		

*p*—statistical significance χ^2^ test.

**Table 2 ijerph-19-10915-t002:** Frequency of genotypes of the *DRD2* gene rs1799732 gene polymorphisms in the Cannabis Dependency and control subjects.

	*DRD2* rs1799732				
	Genotypes			Alleles	
	Del/Del n (%)	Ins/Ins n (%)	Ins/Del n (%)	Del n (%)	Ins n (%)
Cannabis Dependency n = 214	7 (3.27%)	158 (73.83%)	49 (22.90%)	63 (14.72%)	365 (85.28%)
Control n = 301	4 (1.33%)	241 (80.07%)	56 (18.60%)	64 (10.63%)	538 (89.37%)
χ^2^ (*p* value)	3.966 0.138			3.870 (0.049) *	

n—number of subjects. *—significant statistical differences.

**Table 3 ijerph-19-10915-t003:** STAI and NEO Five-Factor Inventory sten scores between healthy controls and Cannabis Dependency.

STAI/NEO Five-Factor Inventory	Cannabis Dependency (n = 214)	Control (n = 301)	Z	(*p*-Value)
STAI trait/scale	7.10 ± 2.35	5.16 ± 2.18	8.619	0.0000 *
STAI state/scale	5.82 ± 2.44	4.69 ± 2.14	5.418	0.0000 *
Neuroticism/scale	6.70 ± 2.25	4.68 ± 2.02	9.472	0.0000 *
Extraversion/scale	5.66 ± 2.15	6.37 ± 1.97	−3.667	0.0002 *
Openness/scale	5.04 ± 2.00	4.53 ± 1.61	2.835	0.0045 *
Agreeability/scale	4.29 ± 1.97	5.60 ± 2.09	−6.825	0.0000 *
Conscientiousness/scale	5.49 ± 2.25	6.08 ± 2.15	−2.845	0.0044 *

*p*, statistical significance with Mann–Whitney U-test; n, number of subjects; M ± SD, mean ± standard deviation; * statistically significant differences.

**Table 4 ijerph-19-10915-t004:** Differences in the *DRD2* gene rs1799732 and NEO Five-Factor Inventory, STAI scale between healthy control subjects and cannabinoid dependence.

STAI/NEO Five-Factor Inventory	Group	*DRD2* Gene rs1799732				ANOVA		
		Del/Del n = 11 M ± SD	Ins/Ins n = 397 M ± SD	Ins/Del n = 104 M ± SD	Factor	F (*p* Value)	ɳ^2^	Power (Alfa = 0.05)
STAI trait/scale	Cannabinoid dependence (CD); n = 214	7.57 ± 1.81	6.95 ± 2.33	7.52 ± 2.45	Intercept CD/control *DRD2* CD/control × *DRD2*	F_1,507_ = 597.45 (*p* < 0.0001) F_1,507_ = 27.02 (*p* < 0.0001) F_2,507_ = 0.16 (*p* = 0.844) F_2,507_ = 4.39 (*p* = 0.013) *	0.541 0.051 0.001 0.017	1.000 0.999 0.076 0.757
	Control; n = 301	4.50 ± 1.91	5.33 ± 2.16	4.48 ± 2.17				
STAI state/scale	Cannabinoid dependence (CD); n = 214	6.57 ± 2.15	5.71 ± 2.41	6.08 ± 2.62	Intercept CD/control *DRD2* CD/control × *DRD2*	F_1,507_ = 430.42 (*p* < 0.0001) F_1,507_ = 12.60 (*p* < 0.001) F_2,507_ = 0.04 (*p* = 0.962) F_2,507_ = 1.48 (*p* = 0.228)	0.459 0.024 0.0001 0.006	1.000 0.943 0.056 0.317
	C: Control; n = 301	3.75 ± 1.50	4.75 ± 2.18	4.50 ± 2.02				
Neuroticism/scale	Cannabinoid dependence (CD); n = 214	6.85 ± 1.77	6.61 ± 2.19	6.98 ± 2.50	Intercept CD/control *DRD2* CD/control × *DRD2*	F_1,507_ = 546.67 (*p* < 0.0001) F_1,507_ = 32.57 (*p* < 0.0001) F_2,507_ = 0.45 (*p* = 0.637) F_2,507_ = 1.90 (*p* = 0.151)	0.519 0.060 0.002 0.007	1.000 0.999 0.123 0.395
	Control; n = 301	3.25 ± 2.06	4.76 ± 2.01	4.41 ± 2.02				
Extraversion/scale	Cannabinoid dependence (CD); n = 214	4.28 ± 2.43	5.65 ± 2.17	5.87 ± 2.03	Intercept CD/control *DRD2* CD/control × *DRD2*	F_1,507_ = 717.55 (*p* < 0.0001) F_1,507_ = 12.53 (*p* = 0.0004) F_2,507_ = 0.59 (*p* = 0.553) F_2,507_ = 2.34 (*p* = 0.097)	0.586 0.024 0.002 0.009	1.000 0.942 0.148 0.474
	Control; n = 301	7.75 ± 0.50	6.30 ± 1.98	6.57 ± 1.98				
Openness/scale	Cannabinoid dependence (CD); n = 214	4.57 ± 2.23	4.95 ± 2.02	5.37 ± 1.91	Intercept CD/control *DRD2* CD/control × *DRD2*	F_1,507_ = 565.02 (*p* < 0.0001) F_1,507_ = 1.87 (*p* = 0.171) F_2,507_ = 0.62 (*p* = 0.535) F_2,507_ = 0.77 (*p* = 0.465)	0.527 0.004 0.002 0.003	1.000 0.276 0.154 0.180
	Control; n = 301	4.25 ± 2.99	4.55 ± 1.59	4.48 ± 1.61				
Agreeability/scale	Cannabinoid dependence (CD); n = 214	5.00 ± 2.23	4.30 ± 2.04	4.17 ± 1.69	Intercept CD/control *DRD2* CD/control × *DRD2*	F_1,507_ = 583.61 (*p* < 0.0001) F_1,507_ = 17.50 (*p* < 0.0001) F_2,507_ = 4.33 (*p* = 0.013) * F_2,507_ = 1.39 (*p* = 0.249)	0.535 0.033 0.017 0.005	1.000 0.987 0.751 0.299
	Control; n = 301	8.25 ± 2.36	5.64 ± 2.11	5.21 ± 1.85				
Conscientiousness/scale	Cannabinoid dependence (CD); n = 214	6.14 ± 1.34	5.45 ± 2.32	5.54 ± 2.13	Intercept CD/control *DRD2* CD/control × *DRD2*	F_1,507_ = 629.99 (*p* < 0.0001) F_1,507_ = 2.61 (*p* = 0.107) F_2,507_ = 1.04 (*p* = 0.351) F_2,507_ = 0.09 (*p* = 0.917)	0.554 0.005 0.004 0.0003	1.000 0.364 0.233 0.063
	Control; n = 301	7.25 ± 2.99	6.02 ± 2.11	6.21 ± 2.25				

*—significant result; CD—cannabinoid dependence; M ± SD—mean ± standard deviation.

**Table 5 ijerph-19-10915-t005:** Post-hoc test (Bonferroni) analysis of interactions between the patients diagnosed with polysubstance use disorder comorbid with cannabinoid dependence/control and the *DRD2* gene rs1799732 and STAI trait scale.

*DRD2* Gene rs1799732 and STAI Trait Scale						
	{1} M = 7.57	{2} M = 6.96	{3} M = 7.52	{4} M = 4.50	{5} M = 5.33	{6} M = 4.48
Cannabinoid dependence del/del {1}		1.0000	1.0000	0.4344	0.1396	0.0093 *
Cannabinoid dependence ins/ins {2}			1.0000	0.4604	0.0000 *	0.0000 *
Cannabinoid dependence ins/del {3}				0.1462	0.0000 *	0.0000 *
Control del/del {4}					1.0000	1.0000
Control ins/ins {5}						0.1611
Control ins/del {6}						

*—significant statistical differences; M—mean.

## Data Availability

Not applicable.

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
