# Peer review of "Association between Polymorphism rs1799732 of DRD2 Dopamine Receptor Gene and Personality Traits among Cannabis Dependency"

_ijerph, 2022, doi:10.3390/ijerph191710915_

Round 1
Reviewer 1 Report
I have some comments to be considered by the authors:
INTRODUCTION :
- expand the part about the DRD2 gene and its neurobiological role in cannabis addiction. Some references, such as "Ferland, J. M. N., & Hurd, Y. L. (2020). Deconstructing the neurobiology of cannabis use disorder. Nature Neuroscience, 23(5), 600-610" could be used.
-explain the choice of studying the rs1799732 polymorphism of DRD2 (previous studies showing its role in drug addiction).
Additional remarks :
- the article needs a thorough English editing
-please correct: dependency instead of de-pendency (line 21), and dopamine instead of dopa-mine (title)
-Line 134 : please remove the sentence « interventionary studies involving animals or 134 humans, and other studies that require ethical approval, must list the authority that pro- 135 vided approval and the corresponding ethical approval code ».
-line 154 : correct « table 1 » instead of « table 2 »
Author Response
Dear Reviewer,
Thank you very much for your review and valuable comments. We analyzed all the comments and replied to each, indicating where and how the corrections in the Manuscript were made, indicating the line and page.
Below are the point-by-point answers.
With respect
Authors
INTRODUCTION :
- expand the part about the DRD2 gene and its neurobiological role in cannabis addiction. Some references, such as "Ferland, J. M. N., & Hurd, Y. L. (2020). Deconstructing the neurobiology of cannabis use disorder. Nature Neuroscience, 23(5), 600-610" could be used.
Thanks for your suggestion. The fragment was added on line 57-68 page 2.
-explain the choice of studying the rs1799732 polymorphism of DRD2 (previous studies showing its role in drug addiction).
Thank You. We did our best to correct it. We hope that now it sound better
‘Previous publications have described the relationship of the rs1799732 polymor-phism of DRD2 with nicotine [28] alcohol addiction [29,30,31,32] opioid dependence [33]. Therefore, in our research, we decided to check the relationship of the rs1799732 polymorphism of DRD2 with Cannabis Dependency and with Personality Traits.’
Additional remarks :
- the article needs a thorough English editing
Thank You. Of couese You are write. We did our best to correct it. We hope that now it sound better.
-please correct: dependency instead of de-pendency (line 21), and dopamine instead of dopa-mine (title)
It is done.
-Line 134 : please remove the sentence « interventionary studies involving animals or 134 humans, and other studies that require ethical approval, must list the authority that pro- 135 vided approval and the corresponding ethical approval code ».
It was rejected after Your suggestion.
-line 154 : correct « table 1 » instead of « table 2 »
Of course, it was a typewriting mistake. It’s corrected now.
Reviewer 2 Report
Reviewer’s comment to the author:
In the manuscript entitled “Association between Polymorphism rs1799732 of DRD2 Dopa-mine Receptor Gene and Personality Traits among Cannabis Dependency” authors have shown association between between the rs 1799732 polymorphism of the DRD2 gene and psychological traits and anxiety. In this study, Authors have also investigated the effect of addiction to cannabinoids on the personality traits and anxiety. Though the findings of this manuscript are interesting; however, there are other serious concerns that need attention and require to be discussed.
1. Authors have shown positive correlation between Polymorphism rs1799732 of DRD2 Dopa-mine Receptor Gene and Personality Traits. They should also evaluate how it performs as an independent factor in the presence of other known factors/gene polymorphisms which also influence personality traits in cannabis addiction. This is important as it has been shown earlier that there are many other gene polymorphisms/DRD2 polymorphisms which also have influence on the personality trait and the Cannabis addiction.
2. The authors should also discuss about other known factors and other gene known polymorphisms which affect the personality trait among Cannabis addicted group.
Author Response
Dear Reviewer,
Thank you very much for your review and valuable comments. We analyzed all the comments and replied to each, indicating where and how the corrections in the Manuscript were made, indicating the line and page.
Below are the point-by-point answers.
With respect
Authors
In the manuscript entitled “Association between Polymorphism rs1799732 of DRD2 Dopa-mine Receptor Gene and Personality Traits among Cannabis Dependency” authors have shown association between between the rs 1799732 polymorphism of the DRD2 gene and psychological traits and anxiety. In this study, Authors have also investigated the effect of addiction to cannabinoids on the personality traits and anxiety. Though the findings of this manuscript are interesting; however, there are other serious concerns that need attention and require to be discussed.
- Authors have shown positive correlation between Polymorphism rs1799732 of DRD2 Dopa-mine Receptor Gene and Personality Traits. They should also evaluate how it performs as an independent factor in the presence of other known factors/gene polymorphisms which also influence personality traits in cannabis addiction. This is important as it has been shown earlier that there are many other gene polymorphisms/DRD2 polymorphisms which also have influence on the personality trait and the Cannabis addiction.
- The authors should also discuss about other known factors and other gene known polymorphisms which affect the personality trait among Cannabis addicted group.
Thank You for these notes. We really too scanty described other gene polymorphisms which are known to affect personality traits in part of discussion our our results. We tried to bind together Your both suggestions and we tried to fulfill it in thecompact and satisfactory way. We didn’t described any other known gene polymorphisms (e.g.COMT, DAT, 5-HTTLPR) which could be connected with cannabis use disorder - except the examinated DRD2 gene - and their relationships with personality traits. As we considered that it could have been to unclear and “overdosed” in this one publication. Against of it, we hope that You will find the discussion more interesting and complet now.
Reviewer 3 Report
The manuscript entitled "Association between Polymorphism rs1799732 of DRD2 Dopamine Receptor Gene and Personality Traits among Cannabis Dependency" presents a classic approach to evaluating genetic susceptibility to drug dependency.
Particularly, I believe the genetic analyses is well-described and performed. I have a few considerations for the authors:
1. DRD2 gene function is described correctly, but I missed an explanation regarding the polymorphism in the Introduction. Is it functional? What is the impact on the regulatory processes since it is an intronic variant?
2. In the same idea, this polymorphism has been associated with several psychiatric conditions. I missed addressing these other conditions in the Discussion section (only in the Ensembl database there are 185 citations comprising rs1799732).
3. In the Material & Methods section, the authors mention the genotyping details have been described previously (line 132), but no reference is provided.
4. In the Discussion section, the authors summarize the results of the NEO-FFI inventory (lines 249-251) but present only the same conclusion already mentioned in the Results. I think the implication of this result should be better detailed, especially for the readers that are not familiar with this inventory.
5. Minor point: sometimes the polymorphism is cited as rs179973, missing the last number.
Author Response
Dear Reviewer,
Thank you very much for your review and valuable comments. We analyzed all the comments and replied to each, indicating where and how the corrections in the Manuscript were made, indicating the line and page.
Below are the point-by-point answers.
With respect
Authors
The manuscript entitled "Association between Polymorphism rs1799732 of DRD2 Dopamine Receptor Gene and Personality Traits among Cannabis Dependency" presents a classic approach to evaluating genetic susceptibility to drug dependency.
Particularly, I believe the genetic analyses is well-described and performed. I have a few considerations for the authors:
- DRD2 gene function is described correctly, but I missed an explanation regarding the polymorphism in the Introduction. Is it functional? What is the impact on the regulatory processes since it is an intronic variant?
Thank You. We did our best to correct it.
‘One DRD2 gene polymorphism that has been investigated extensively is a cytosine (C) insertion/deletion (Ins/Del) at nucleotide position −141C (−141C Ins/Del, rs1799732) in the promoter region, which may regulate DRD2 transcription by modulating the binding of transcription factors [28]. Arinami et al. [28] show that variation in the genomic sequence of the promoter region of the D2 receptor gene (DRD2) could affect the expression or regulation of the gene. The DRD2 5′-promoter fragments drove the transcription of heterologous luciferase constructs in Y79 cell line expressing DRD2 as well as in DRD2 non-expressing 293 cells. The fragment that contained the −141C Del allele showed a decrease in promoter strength as compared with the fragment that contained the −141CIns allele in Y-79 and 293 cells. The position of the polymorphism is part of a putative binding site for the transcription factor Sp-1, 5′-CCAGGCCGGGGATCGCC.In vivo study, Jönsson et al. [29] found a significant association between the presence of a putative functional DRD2 promoter allele (−141C Del) and high striatal DA receptor density in healthy subjects. These studies suggested that the rs1799732 polymorphism may be involved in the regulation of DRD2 expression (mRNA level and/or protein level)’.
- In the same idea, this polymorphism has been associated with several psychiatric conditions. I missed addressing these other conditions in the Discussion section (only in the Ensembl database there are 185 citations comprising rs1799732).
Thank You for this suggestion, though we unfortunally didn’t manage to find any publications which could be exactly related to DRD2 rs1799732 polymorphism with affective diseases, general anxiety disorder, posttraumatic stress disorder only in schizophrenia, it treatment and tardive dyskinesia. These last were not a point of our interesting in this matter. These articles we got were placed in the discussion. We hope that now it sounds much better and complete.
- In the Material & Methods section, the authors mention the genotyping details have been described previously (line 132), but no reference is provided.
Thank You for this suggestion. It corrected now.
- In the Discussion section, the authors summarize the results of the NEO-FFI inventory (lines 249-251) but present only the same conclusion already mentioned in the Results. I think the implication of this result should be better detailed, especially for the readers that are not familiar with this inventory.
Thank You for this observation. We tried to rebuilt once again the discussion part to fulfill it with some new information which could be helpful in understanding this complex psychological and psychiatric problems connected with the main idea of drug/cannabis dependency. We are consious of the difficulties in catching these relationships and therefore we tried to explained them as simply as we managed. We hope it will be acceptable now.
- Minor point: sometimes the polymorphism is cited as rs179973, missing the last number.
It corrected now.